# Piscine Orthoreovirus (PRV)-3, but Not PRV-2, Cross-Protects against PRV-1 and Heart and Skeletal Muscle Inflammation in Atlantic Salmon

**DOI:** 10.3390/vaccines9030230

**Published:** 2021-03-06

**Authors:** Muhammad Salman Malik, Lena H. Teige, Stine Braaen, Anne Berit Olsen, Monica Nordberg, Marit M. Amundsen, Kannimuthu Dhamotharan, Steingrim Svenning, Eva Stina Edholm, Tomokazu Takano, Jorunn B. Jørgensen, Øystein Wessel, Espen Rimstad, Maria K. Dahle

**Affiliations:** 1Faculty of Veterinary Medicine, Norwegian University of Life Sciences, 0454 Oslo, Norway; muhammad.salman.malik@nmbu.no (M.S.M.); l.h.teige@medisin.uio.no (L.H.T.); stine.braaen@nmbu.no (S.B.); dhamotharan.kannimuthu@hi.no (K.D.); oystein.wessel@nmbu.no (Ø.W.); espen.rimstad@nmbu.no (E.R.); 2Department of Fish Health, Norwegian Veterinary Institute, 0454 Oslo, Norway; anne-berit.olsen@vetinst.no (A.B.O.); marit.masoy.Amundsen@vetinst.no (M.M.A.); 3Norwegian College of Fishery Science, Faculty of Biosciences, Fisheries and Economics, UiT The Arctic University of Norway, 9019 Tromsø, Norway; mno085@post.uit.no (M.N.); steingrim.svenning@uit.no (S.S.); eva-stina.i.edholm@uit.no (E.S.E.); jorunn.jorgensen@uit.no (J.B.J.); 4National Research Institute of Aquaculture, Japan Fisheries Research and Education Agency, Nansei 516-0193, Japan; takanoto@fra.affrc.go.jp

**Keywords:** heart and skeletal muscle inflammation, *Piscine orthoreovirus*, vaccine, atlantic salmon, antibodies, immune response

## Abstract

Heart and skeletal muscle inflammation (HSMI), caused by infection with *Piscine orthoreovirus-1* (PRV-1), is a common disease in farmed Atlantic salmon (*Salmo salar*). Both an inactivated whole virus vaccine and a DNA vaccine have previously been tested experimentally against HSMI and demonstrated to give partial but not full protection. To understand the mechanisms involved in protection against HSMI and evaluate the potential of live attenuated vaccine strategies, we set up a cross-protection experiment using PRV genotypes not associated with disease development in Atlantic salmon. The three known genotypes of PRV differ in their preference of salmonid host species. The main target species for PRV-1 is Atlantic salmon. Coho salmon (*Oncorhynchus kisutch*) is the target species for PRV-2, where the infection may induce erythrocytic inclusion body syndrome (EIBS). PRV-3 is associated with heart pathology and anemia in rainbow trout, but brown trout (*S. trutta*) is the likely natural main host species. Here, we tested if primary infection with PRV-2 or PRV-3 in Atlantic salmon could induce protection against secondary PRV-1 infection, in comparison with an adjuvanted, inactivated PRV-1 vaccine. Viral kinetics, production of cross-reactive antibodies, and protection against HSMI were studied. PRV-3, and to a low extent PRV-2, induced antibodies cross-reacting with the PRV-1 σ1 protein, whereas no specific antibodies were detected after vaccination with inactivated PRV-1. Ten weeks after immunization, the fish were challenged through cohabitation with PRV-1-infected shedder fish. A primary PRV-3 infection completely blocked PRV-1 infection, while PRV-2 only reduced PRV-1 infection levels and the severity of HSMI pathology in a few individuals. This study indicates that infection with non-pathogenic, replicating PRV could be a future strategy to protect farmed salmon from HSMI.

## 1. Introduction

Infections represent a constant challenge and threat against fish health and welfare in aquaculture. Modern farming of Atlantic salmon (*Salmo salar*) is characterized by high-density populations, rapid growth, short production cycles, and artificial adaptation to sea water. This life cycle does not ensure natural pathogen exposure in early life or the natural training of the fish innate immune system [1]. When transferred to the sea, the untrained immune system may not be ready to handle the novel repertoire of pathogens. High-density populations increase infection pressure, and transportation and handling procedures increase disease susceptibility due to stress [2]. In Atlantic salmon aquaculture, vaccines have been effective in protecting the fish from many diseases, but several viral diseases remain unsolved challenges [3]. One of the viral diseases of concern in European Atlantic salmon aquaculture is heart and skeletal muscle inflammation (HSMI) caused by *Piscine orthoreovirus* (PRV) [4,5]. 

PRV particles are non-enveloped with a double-layered protein capsid and a segmented double-stranded RNA genome [6]. PRV is a common virus infection in salmonids, and PRV-1 is the genotype associated with HSMI in farmed Atlantic salmon [5,7]. PRV is ubiquitous in the sea water phase of salmonid aquaculture [8] and is also emerging in fresh water facilities. However, PRV-1 is found to a lower extent in salmonids in the wild [9,10]. PRV-1 was first described in 2010 [4], whereas HSMI emerged in Norway and Scotland a decade earlier [11,12]. The causality between PRV-1 and HSMI was proven experimentally in 2017 using highly purified virus to induce disease [5]. PRV-1 is proposed to infect Atlantic salmon via the intestinal tract [13], followed by a massive infection of red blood cells and high plasma viremia [14,15]. Following the peak infection in red blood cells, the virus infects cardiomyocytes, which may result in an inflammatory response dominated by cytotoxic T-cells in the heart [16,17]. This inflammatory response is a hallmark of HSMI. In Atlantic salmon populations, the disease usually gives a moderate mortality that in severe cases may accumulate to 20% [11]. The relative high frequency of outbreaks makes HSMI a significant problem for the salmon farming industry. The PRV-1 infection becomes persistent in Atlantic salmon, and based on PRV prevalence in farm escapees [10], near 90% of Norwegian farmed salmon are PRV-infected in the marine phase, while near 100% of a small number of escaped Atlantic salmon were reported infected in Washington and British Columbia [18]. The long-term effects of PRV-1 infection are disputed, but the virus has been associated with the worsening of black spots in the skeletal muscle [19], a significant quality problem for the salmon production industry. This association is, however, disputed [20]. PRV-1 is also found in Canadian aquaculture, but few cases of HSMI have been reported [21], and HSMI has not been reproduced experimentally using Canadian isolates [22,23,24]. Different PRV-1 isolates with genetic variation have been shown to differ in the ability to induce HSMI [7]. PRV-1 has also been reported to infect other salmonid species [25]. 

Two additional genotypes of PRV, PRV-2 and PRV-3, have been described. They both infect salmonids, but with a different ability to infect and cause disease in the various salmonid species. PRV-2 infects coho salmon (*Oncorhynchus kisutch*) in Japan, causing erythrocytic inclusion body syndrome (EIBS) [26]. The main host species of PRV-3 may be wild brown trout (*S. trutta*) [27], but disease has only been found in farmed rainbow trout (*O. mykiss*), where PRV-3 is associated with heart inflammation and anemia [28,29,30]. Nucleotide alignment shows 80% (PRV-2) and 89% (PRV-3) identity to PRV-1 [31]. PRV-3 has previously been shown to infect Atlantic salmon experimentally, but without inducing HSMI [29]. Current information on PRV subtypes and distribution was recently reviewed [32].

No vaccines have been marketed against HSMI, but two different experimental vaccination approaches have been published. An inactivated whole virus vaccine, based on purified virus, was shown to give partial protection against HSMI, but less efficient protection against infection and virus replication [33]. Although promising, this approach has been hampered by the problem of producing PRV-1 for vaccine development, as no cell lines efficiently produce viral progeny [34]. A DNA vaccine approach has also been tested, and partial protection against HSMI was reported for a vaccine combining non-structural PRV-1 proteins with outer capsid antigens [35]. Although with some protective effects against HSMI, none of these vaccines have been able to block PRV-1 infection.

PRV-1 infection has been reported to induce strong innate antiviral responses in infected red blood cells [36]. Expression analysis of adaptive immune response genes has indicated that both humoral and cellular responses are induced [37], and it has been shown that infected fish produce specific antibodies against the outer capsid spike protein σ1 [38], predicted to be the receptor-binding protein [39]. The cellular immune response initiated by PRV-1 in Atlantic salmon is strongly associated with HSMI development, and the typical HSMI myocarditis is dominated by an influx of cytotoxic T-cells [16,17]. However, this response is also associated with virus eradication from heart tissue, making cellular immunity a two-edged sword in HSMI [16,40].

The purpose of this study was to determine if PRV-2 or PRV-3 infection in Atlantic salmon could provide protection against a consecutive PRV-1 infection and HSMI. We compared the protection induced by PRV-2 and PRV-3 to an inactivated PRV-1 vaccine, and characterized immune responses, including the production of cross-reactive PRV-specific antibodies. The results show that PRV-3 infection in Atlantic salmon, in contrast to PRV-2, blocks a secondary infection with PRV-1, and that cross-protective antibodies may be one of the mechanism involved.

## 2. Materials and Methods

### 2.1. Experimental Trial and Sampling 

The trial was performed at the Aquaculture Research station at Kårvika, Troms, Norway, approved by the Norwegian Animal Research Authority, and performed in accordance with the recommendations of the current animal welfare regulations: FOR-1996-01-15-23 (Norway).

The PRV-1 infection material was prepared from two frozen blood cell pellets (−80 °C) with PRV-1 qPCR ct values of 17.6 and 16.4, harvested from a PRV-1-infected Atlantic salmon from a previous experimental trial [5]. The virus isolate (PRV-1 NOR2012-V3621 [5]) originated from an HSMI outbreak in mid-Norway in 2012 and had been passaged in prior experimental trials, all resulting in HSMI. The PRV-3 infection material was a blood pellet that originated from a Norwegian outbreak in 2014 (PRV-3 NOR2014, [28]) and has been passaged twice experimentally in rainbow trout [30]. The mock-blood cell lysate originated from control fish from an Atlantic salmon experimental trial. The blood cell lysate from PRV-1, PRV-3 and mock was prepared by diluting the blood pellet (plasma removed prior to freezing) 1:10 in L15-medium, sonicating five times at 20 kHz for 10 s with 1 min rest in between and centrifuging at 3000× *g* for 10 min before the collection of the supernatant. The PRV-2 infection material (PRV-2, [26]) originated from a frozen spleen sample from a Coho salmon. The tissue sample was homogenized in L15 medium as described for the blood pellets. The inactivated PRV-1 material was prepared from a batch of purified PRV-1 particles (PRV-1 NOR2012, 5.35 × 10^9^ copies /mL) by PHARMAQ AS, as described in a previously published trial [33]. In short, the batch was formalin-inactivated and prepared as a water-in-oil formulation where the water phase (containing PRV antigens) was dispersed into a mineral oil continuous phase containing emulsifiers and stabilizers. 

At the start of the trial, a total of 630 fish (*Salmo salar* L) were divided into four experimental groups of 75 fish and one mock control group of 125 fish, while 190 naïve fish from the same group were kept for use as transmission controls and future virus shedders. The experimental fish were kept in freshwater (10 °C, 24:0 light:dark cycle, >90% O_2_) and injected intraperitoneally (ip) with 0.2 mL of immunization material described above. Eight fish were sampled prior to Injection Week 0, and from each of the five experimental groups Week 2 and 5. Five weeks after the start of the experiment, 12 naïve fish labelled by tattoo pen were added to each of the tanks containing fish infected with PRV-1, PRV-2 and PRV-3 to monitor transmission of virus. At Weeks 8 and 10, eight experimental fish and six transmission control fish were sampled from each of these groups. At Week 8, 140 naïve fish in a separate tank were injected ip with 0.2 mL of a newly prepared batch of PRV-1 blood cell lysate (PRV-1 NOR2012, same origin and preparation method) and left for two weeks. After Sampling Week 10, 35 fish remained in each of the experimental groups, and 70 fish in the mock-infected control group. The mock group fish were divided into two tanks of 35 fish each, and three experimental tanks (PRV-2, PRV-3, InactPRV-1) and one of the mock-tanks were added to an equal number (35) of tattoo-labelled pre-infected PRV-1 shedder fish. No shedders were added to the original PRV-1 tank, and the other mock group was kept as a negative control. The number of tanks included in the experiment was now 6, and eight fish from each group were sampled on Weeks 12, 15 and 18. No fish died during the experiment. 

At each sampling, blood was drawn from the caudal vein on BD Medical Vacutainer heparin-coated tubes (BD Medical, Mississauga, ON, USA). Hearts were sampled on 10% formalin for histology and samples from the heart tip and spleen were sampled on 0.5 mL of RNALater (Qiagen, Hilden, Germany) in separate bar-coded microtubes (FluidX Ltd., Manchester, UK) along with additional organ samples not analyzed here. Blood samples were stored at 4 °C for a maximum of 6 h, centrifuged (3000× *g* for 5 min at 4 °C), and plasma and cell pellets were separated into different microtubes and stored at −80 °C. RNALater samples were stored at 4 °C for 24 h followed by freezing at −20 °C. Formalin samples were stored at RT for 24 h, after which formalin was changed to 70% ethanol, and thereafter stored cold (4 °C).

### 2.2. RNA Preparation and RT-qPCR for Virus and Host Response Gene Analyses

Tissue samples from the spleen and heart (25 mg) on RNALater (Qiagen) were transferred to 0.65 mL Qiazol lysis reagent (Qiagen) with a 5 mm steel bead and homogenized in a TissueLyzer II (Qiagen) for 2 × 5 min at 25 Hz followed by chloroform inclusion, and the aqueous phase was collected. RNeasy Mini QIAcube Kit (Qiagen) was used as per the manufacturer guidelines for automated RNA isolation. RNA concentrations were quantified using the Nanodrop ND-100 spectrophotometer (Thermo Fisher Scientific, Waltham, MA, USA). RNA was eluted in RNase-free water and stored at −80 °C until further use. For the PRV subtype expression analysis, i.e., PRV-1 and PRV-3, one-step RT-qPCR was performed using an Agilent Brilliant III Ultra Fast kit (Agilent Technologies, Santa Clara, CA, USA) with 100 ng (5 µL of 20 ng/µL) RNA per reaction in duplicates of 15 µL total reaction volume. The template was previously denatured at 95 °C for 5 min. Cycling parameters were set to 10 min for 50 °C, 3 min at 95 °C, and 40 cycles for 5 s at 95 °C and 10 s at 60 °C. The cut-off value was set to 35 and samples were run with positive and no template controls (NTC). For PRV-2 expression analysis, a Quantitect SYBR Green (Qiagen) RT-qPCR kit (catalogue number 204243) was used according to manufacturer instructions. A total of 100 ng RNA with prior denaturation at 95 °C for 5 min was used in duplicates in 15 µL of total reaction volume. Thermal conditions were 50 °C for 30 min, 95 °C for 15 min, and 40 cycles with 94 °C for 15 s, 60 °C for 30 s and 72 °C for 30 s. Specificity of the assay was confirmed by melting curve analysis. The same threshold level and positive controls were used together with NTCs. Probes and primer sequences are given in Appendix A. 

For Immune gene expression, 400 ng total spleen RNA per sample was reverse transcribed to cDNA using the QuantiTect Reverse Transcription Kit with gDNA wipeout buffer (Qiagen). For qPCR, cDNA corresponding to 5 ng RNA was analyzed with Sso Advanced Universal SYBR Green Supermix (Bio-Rad, Hercules, CA, USA) and 10 pmol of forward and reverse target-specific primers in a 10 µL volume in duplicate wells on a 384 well plate. The amplification program (15 s 95 °C, 30 s 60 °C) was run for 40 cycles in a CFX Touch Real-Time PCR Detection System (Bio-Rad), followed by a melt point analysis. The results were analyzed using the software CFX Manager, version 3.1.1621.0826. The expression cycle threshold level was normalized against the elongation factor (EF) 1α reference gene (ΔCt). The ΔΔCt method was used to calculate relative expression levels and fold induction compared to samples from the uninfected control samples.

### 2.3. Bead-Based Immunoassay

MagPlex^®^-C Microspheres (Luminex Corp., Austin, TX, USA) #12, #21, #27, #29, #34, #36, #44, #62 and #64 were coated with antigens using the Bio-Plex Amine Coupling Kit (Bio-Rad, Hercules, CA, USA) according to the manufacturer’s instructions. The N-Hydroxysulfosuccinimide sodium salt and N-(3-Dimethylaminopropyl)-N’-ethylcarbod used for the coupling reaction were from Sigma-Aldrich. For each coupling reaction, 6-24 μg of recombinant protein was used. The proteins used were recombinant PRV μ1l [41], lipid-modified PRV σ1 (LM-PRVσ1), unmodified infectious salmon anemia virus fusion protein (ISAV-FP), and lipid-modified ISAV-FP (LM-ISAV-FP) [39]. The bead concentrations were determined using the Countess automated cell counter (Invitrogen, Carlsbad, CA, USA). Coupled beads were stored in black Eppendorf tubes at 4 °C for up to 10 weeks. Incubations were performed at room temperature and protected from light on a HulaMixer rotator (Thermo Fisher Scientific) at 15 rpm.

The immunoassay was performed as described earlier (8). Briefly, Bio-Plex Pro™ Flat Bottom Plates (Bio-Rad) were used. Beads were diluted in phosphate buffered saline (PBS) containing 0.5% bovine serum albumin (BSA) (Bio-Rad Diagnostics GmbH, Dreieich, Germany) and 0.05% azide (Merck, Darmstadt, Germany) (PBS+) and 2500 beads of each bead number were added to each well. AntiSalmonid-IgH monoclonal antibody (clone IPA5F12, Cedarlane, Burlington, ON, Canada) diluted 1:400 in PBS+ was used as an unconjugated anti-IgM heavy chain monoclonal antibody. Biotinylated goat AntiMouse IgG2a antibody (Southern Biotechnology Association, Birmingham, AL, USA) diluted 1:1000 in PBS+ was used as a secondary antibody, and Streptavidin-PE (Invitrogen) diluted 1:50 in PBS+ was used as the reporter flourochrome. Plates were read using two different Bio-Plex 200 (Bio-Rad) machines as part of a validation plan. The DD-gate was set to 5000–25,000, and between 20 and 100 beads from each population were read from each well. The reading was carried out using a low (standard) photomultiplier tube (PMT) setting. The results were analyzed using the Bio-Plex Manager 5.0 and 6.1 (Bio-Rad). All samples were analyzed in duplets on each of the two different Bio-Plex 200 (Bio-Rad) machines. The data used originated from one machine, but no differences were observed during validation. The data were given in mean fluorescence intensity (MFI), based on secondary antibody binding to beads, and were corrected for binding to control beads without antigen: MFI (antigen-containing beads) − MFI (control beads) = MFI (sample data).

### 2.4. Histopathology

Formalin-fixed hearts were paraffin embedded and routinely processed. The sections, 3–4 μm, were stained with hematoxylin and eosin (H&E, Merck, Kenilworth, NJ, USA) and studied under microscope. The slides from Experimental Weeks 15 and 18 (*n* = 96) were blinded to the study groups and scored by an experienced fish pathologist using a visual analogue scale from 0 to 3 as previously described [11].

### 2.5. Statistical Analysis 

All statistical analyses were performed within GraphPad Prism 8.1.1 (GraphPad Software Inc., La Jolla, CA, USA). Ct values of the target groups (PRV-2, PRV-3 and Inact. PRV-1-injected fish exposed to PRV-1 shedder fish at 10 weeks post injection) were compared to the PRV-1 control group by using the non-parametric Mann–Whitney test due to the small sample size (*n* = 8) in each group. *p*-values of *p* ≤ 0.05 were considered as significant.

## 3. Results

### 3.1. PRV Immunization Trial

The trial was performed as outlined in Figure 1. Initially (Week 0), Atlantic salmon with a mean weight of 41.3 g (+/− 5.8 g) were grouped and injected intraperitoneally (ip) with cell or tissue lysates containing infective PRV-1, PRV-2 or PRV-3, uninfected blood lysate (mock), or purified, inactivated and adjuvanted PRV-1 [33]. At 10 weeks, the mean weight of the injected fish was 107.6 g (+/− 18.4 g) with no significant difference between groups (Appendix A). At this timepoint, PRV-1-infected shedder fish were added to the remaining fish in the groups injected with PRV-2, PRV-3, inactivated PRV and half of the mock group to test the effects of immunization. Neither the initial ip challenge/immunization nor the cohabitant challenge led to mortality in any of the treatment groups, and there was no loss of fish or aberrant clinical observations during the experimental period. At the end of the experiment in Week 18, the fish mean weight was 193.6 g (+/− 29.5 g), with no statistically significant difference between groups. 

### 3.2. Replication and Transmission of PRV Genotypes in Atlantic Salmon

The RNA loads of PRV-1, PRV-2 and PRV-3 were monitored by the RT-qPCR of spleen samples through the experimental period (Figure 2, Appendix A). The spleen was chosen for analysis since PRV replicates in red blood cells, and spleen has been shown to reflect the levels of PRV infection in blood [42] better than, e.g., kidney. PRV-1 showed maximum replication during the first 5 weeks, as expected from previous trials (median Ct 14.79, interquartile range (IQR) Ct 14.12–15.37 (Figure 2A)), and persisted in spleen through the 18 weeks of the study with median PRV-1 levels above a Ct level of 20 at all sampling time points. Five weeks after injection, naïve fish were added to tanks of fish injected with PRV-1, PRV-2 and PRV-3 to study the transmission of the injected virus. The naïve cohabitants added to the PRV-1 group at Week 5 were all infected 3 and 5 weeks later (Experimental Week 8 and 10, not analyzed at later time points). PRV-2 levels were generally low and reached the highest level after 2 weeks (median Ct of 26.7, IQR Ct 25.99–27.08), after which the infection declined. After 18 weeks, PRV-2 was detected in only one out of eight sampled fish. No naïve cohabitants added to the PRV-2 tank Week 5 were infected (Figure 2B). PRV-3 levels increased up to Week 5 (median Ct of 19.19, IQR Ct 18.02–20.75), then declined until Week 18 (Figure 2C). The added naïve cohabitants were not infected. No cross-infection was observed between the tanks, and no replication was observed in the fish injected with inactivated PRV-1, as monitored on Weeks 2, 5 and 10 (Appendix A, Appendix A). 

To explore if there was any persistence of PRV2 and PRV3 in hearts at the end of the trial, we compared RNA loads of PRV-1, PRV-2 and PRV-3 in heart samples at 15 and 18 weeks (Figure 2D). Whereas PRV-1 levels in heart were below Ct 25, PRV-2 was only detected (median 34.87, IQR Ct 34.31–37.24) in the heart in two fish at 15 weeks after infection, and one fish at 18 weeks. PRV-3 was detected at low levels in 50% of the fish hearts at both time points. Except for two fish at 15 weeks, all PRV-3-positive fish had Ct levels above 30 in the heart. 

### 3.3. Production of Anti-PRV Antibodies 

Using a bead-based multiplexed immunoassay based on recombinant PRV-1 spike protein σ1 and outer capsid protein µ1c [39], the ability of the viruses to induce cross-binding antibodies in plasma (IgM) was explored for the period 2 to 10 weeks after virus injection (Figure 3, Appendix A). PRV-1 infection induced the production of PRV-1-specific antibodies against the viral proteins σ1 and µ1 after 8 and 10 weeks (Figure 3A) and induced unspecific antibodies binding to non-PRV antigens. PRV-2 induced low levels of PRV-1 σ1 binding antibodies as detected at Weeks 5 and 8, declining at Week 10 (Figure 3B), in line with a low PRV-2 replication in the fish. PRV-3 infection induced intermediate levels of PRV-1 σ1 binding antibodies, with lower background binding to non-PRV antigens (Figure 3C). Inactivated PRV-1 did not induce detectable production of antibodies binding to PRV-1 σ1 (Figure 3D). 

### 3.4. Innate and Cellular Immune Responses 

In order to explore which immune responses were activated in the fish at the time of exposure to PRV-1 shedder fish (10wpi), spleen RNA samples were analyzed for transcript markers of cellular cytotoxic immunity (Figure 4, Appendix A): CD8α, IFN-γ and Granzyme A (Figure 4A), and innate interferon-mediated antiviral responses: viperin, myxovirus resistance gene (Mx), and interferon-stimulated gene (ISG)15 (Figure 4B). These genes have previously been shown to be induced in spleen after infection with PRV-1 [37]. PRV-1 infection induced both cellular and innate immune responses in spleen, whereas infection with PRV-2, PRV-3 or inactivated adjuvanted PRV-1 showed no or minor induction of the cellular and selected innate antiviral response genes. 

### 3.5. Protection from PRV Infection and HSMI

Infection with PRV-1 was monitored in all groups from 12 to 18 wpi (Figure 5A,B, Appendix A). The mock-injected + PRV-1-exposed group acted as a positive control and was infected with PRV-1 after two weeks, peaking 5 weeks later (Experimental Week 15) at median Ct levels of 15 in the spleen and median Ct levels of 17 in the heart. Fish that had been immunized with PRV-2 showed a delayed and variable PRV-1 infection level at 15 and 18 weeks ranging from Ct 15 to 30 in the heart and Ct 10 to 24 in the spleen. Surprisingly, the highest PRV-1 infection levels in the PRV-2 group ranged beyond the levels in the positive controls, indicating that PRV-2 increased susceptibility to PRV-1 infection in some individuals. A similar partial protection was seen in the fish immunized with inactivated, adjuvanted PRV-1 (InactPRV-1), but without the replication boost seen in some fish in the PRV-2 group. In fish infected with PRV-3, the PRV-1 infection was completely blocked, except for two individuals showing high Ct levels in the spleen, one of which also had detectable PRV-3 in the heart. 

Hearts from fish sampled at 15 and 18 weeks after PRV-1 infection by shedders, and the corresponding uninfected control group, were prepared for histopathology and scored for tissue changes consistent with HSMI (score system 0–3 [11], Figure 6, Appendix A). At 15 weeks, heart pathology was seen only in the PRV-1 group infected ip at the beginning of the trial (five of eight fish had mild lesions, i.e., a score of 1). HSMI-like lesions were present in all individuals in the mock + PRV-1 control group (positive control) at Week 18, with a median HSMI pathology score of 2.5 (1.5–2.5). For the PRV-2 + PRV-1 group, the median pathology score was reduced to 2 (six out of eight fish had lesions), and for the PRV-3 + PRV-1 group pathology was completely absent in all eight fish (a score of 0). Six out of eight fish from the InactPRV-1 + PRV-1 group were also without heart lesions. The group infected with PRV-1 ip Week 0 showed a median pathology score of 1 (six out of eight fish had mild lesions), 18 weeks after infection.

### 3.6. Immune Responses after Challenge of Immunized Salmon

The specific antibody response (Figure 7A, Appendix A) and cellular cytotoxic immune gene activation—Granzyme A, IFNγ (Figure 7B,C, Appendix A)—were monitored after the PRV-1 challenge at Experimental Weeks 12–18 (two to eight weeks after exposure to shedder fish). The positive control group showed specific and unspecific antibody production and induction of Granzyme A and IFNγ levels in the spleen. The PRV-1-induced antibody response tended to be higher in some fish in the PRV-2 immunized group and lower in fish immunized with inactivated PRV-1. Both observations were in line with the PRV-1 levels found in the spleen. Both groups induced Granzyme A and IFNγ transcripts in fish with high PRV-1 loads, but not in individuals with low PRV-1 loads. In the fish immunized with PRV-3, the antibody levels declined from Week 10 to 18, and since the fish were protected against PRV-1 infection, the antibodies observed most likely resulted from the initial immunization with PRV-3. No regulation of cytotoxic T-cell-associated immune genes was seen. The antibody levels in this group can be compared to the group infected with PRV-1 at Week 0, which showed even higher levels of anti PRV-1 σ1 antibodies during Weeks 12–18. In contrast, whereas fish that were PRV-1 infected Week 0 still had induced levels of Granzyme A in their spleens, the PRV-3-injected group did not.

## 4. Discussion

We clarified the potential of the PRV genotypes PRV-2 and PRV-3 to cross-protect against PRV-1 and HSMI, compared them with an inactivated PRV vaccine, and studied some of the possible protective mechanisms involved. Cross-protection induced by related low virulent virus variants was the first successful immunization strategy more than 200 years ago. It was then found that smallpox was prevented by previous exposure to a low virulent pox virus infecting cows [43]. This strategy was used for several years before it was published by Jenner in 1796. A cross-protective approach to immunization introduces many uncertain factors. The theoretical ability of the low virulent virus to cause low-grade disease, develop into virulence over time, or cause disease in other species requires initial mapping and testing. Nevertheless, a replicating mimic of the disease-causing virus itself has the potential of being the ultimate inducer of efficient immune protection, as this will set off the exact mechanisms used to fight the virus. The rationale for this study is to increase our understanding of cross-protective mechanisms, aiming for the design of more efficient future vaccination approaches. 

Although the three PRV genotypes mainly cause disease in different salmonid species, evidence of cross-species infection exists. PRV-1 infect coho salmon, Chinook salmon, and rainbow trout in addition to Atlantic salmon [25], and PRV-3 infects rainbow trout and coho salmon in addition to brown trout [27]. Our observations of experimental infection of Atlantic salmon with PRV-3 confirmed those of a previous study where it was observed that PRV-3 replicated and persisted over a period of 16 weeks and transmitted less efficiently to naïve cohabitants, compared to PRV-1 [29]. Infection with PRV-2, however, is less studied in other species than farmed coho salmon. Here, we show that PRV-2 can infect and replicate in Atlantic salmon after injection, although not as efficiently as PRV-1 and PRV-3. This ability of both PRV-2 and PRV-3 to infect and replicate in Atlantic salmon calls for awareness of all three viruses in aquaculture and breeding.

As previously shown in several previous experimental challenge studies [5,15,44], the PRV-1 genetic variant used in this trial, originating from a Norwegian disease outbreak, induces HSMI in Atlantic salmon. The same ability to cause HSMI experimentally has not been found for Canadian PRV-1 genetic variants [22,23]. The differences in pathogenicity induced by PRV-1 variants was demonstrated experimentally in 2020 [15], and shown to be associated with genetic differences within four out of the ten genetic segments of PRV. Properties of the outer capsid and virus dissemination in the host was suggested as determinants of pathogenicity [15]. Considering the overall similarities between the PRV genotypes at the amino acid level, PRV-1 is more similar to PRV-3 (90% identity) than to PRV-2 (80%) [31]. The most prominent genetic differences were found in the segment S1, encoding the outer clamp protein σ3 and the non-structural protein p13, encoded by an internal open reading frame. These proteins have both been suggested to be implicated in the pathogenicity of PRV [6,45,46], σ3 for promoting virus replication by dsRNA binding and inhibition of the dsRNA-activated protein kinase PKR [47] and p13 for inducing cytotoxicity [45]. The σ3 and p13 proteins are among the least conserved between the PRV genotypes. For PRV2, σ3 and p13 aa identities to PRV-1 homologues are 69.7 and 62.9%, and for PRV-3 the identities are 79.1 and 78.2%, respectively [31]. The rather low aa conservation could potentially be of importance for the host-specific pathogenicity differences of these viruses, or their ability to interact with each other during infection.

When focusing specifically on the amino acid sequence of the outer capsid protein σ1 (S4) from PRV-1, used as antigen in the bead-based immunoassay [39], the identity is 82% with PRV-3 and only 67% with PRV-2 (NCBI database). Since σ1 is considered to be the receptor-binding protein of PRV [6], its sequence variation may explain the species specificity, and the lack of transmission to naïve cohabitants in Atlantic salmon. The higher amino acid identity between PRV-1 and PRV-3, which is in line with their main host species being more closely related, consequently gave a more efficient infection and replication of PRV-3 compared to PRV-2 in Atlantic salmon. A higher rate of virus replication and higher amino acid identity for σ1 as well as for other virus proteins could explain the higher level of cross-binding antibodies detected after PRV-3 infection and thus the higher cross-protecting effect. Although the genetic diversity in PRV-1 is generally not associated with the σ1 gene, it is possible that cross-protection could be different against the genotypes. 

In this trial, histological analyses were performed only in the late phase of the trial, i.e., after 18 weeks. At that time, PRV-2 was eradicated from the heart, and PRV-3 levels were low, with Ct values above 30 in 50% of the fish and the remaining fish virus being negative. Compared to this, 100% of the fish injected with PRV-1 at the start of the trial were still virus positive in the heart after 18 weeks, with Ct-levels around 20. We cannot completely rule out that heart inflammation occurred at some point after injection with PRV-2 and PRV-3. In a former study on PRV-3 in Atlantic salmon, minor inflammatory foci were detected in the PRV-3-infected hearts [29], but these findings were not comparable, neither to the inflammation induced by PRV-3 in rainbow trout hearts nor to HSMI in Atlantic salmon. Infection and pathological changes in other organs, such as the liver and kidney, earlier shown to be sites for PRV replication [21,48,49], or pathological changes at earlier time points in heart cannot be ruled out either, as this was not explored here. 

Based on analyses of spleen and heart, PRV-2 appears to be eradicated a few weeks after infection in Atlantic salmon compared to PRV-1 and PRV-3. PRV-2 loads in spleen were similar to those of PRV-3 two weeks after infection, but after 5 weeks PRV-2 levels declined, whereas PRV-3 and PRV-1 levels increased. PRV-3 is reported to be successfully cleared in rainbow trout after infection, not moving into persistence like PRV-1 in Atlantic salmon [29,30]. However, PRV-3 appeared to persist for at least 18 weeks in Atlantic salmon in our study, and also for 16 weeks in a former study [29]. This may indicate that persistence is related to host factors in farmed Atlantic salmon. 

In the magnetic bead-based assay used to detect anti-PRV antibodies, the PRV-1 LM-σ1 antigen has earlier been found to be the most efficient antigen for antibody detection [39]. PRV-3 triggered the production of antibodies that were able to cross-bind to PRV-1 LM-σ1. PRV-1 infection has previously been reported to also trigger the production of polyreactive antibodies that bind to non-PRV control antigens [39]. Similarly, we observed high levels of background binding to the ISAV-F-protein control antigens after PRV-1 infection. The production of polyreactive antibodies start at the same time as the more specific antibodies but decrease earlier. The polyreactive antibody response was not seen after PRV-2 or PRV-3 infection. This could be linked to the much higher innate antiviral response triggered by PRV-1, which correlated with the replication efficiency or load of virus for this genotype, compared to the other genotypes. This phenomenon will be subject to further study. 

Low levels of antibodies binding to PRV-1 σ1 was observed in blood from PRV-2-injected individuals as well, but only in a short time frame while the virus was still present. Although this low antibody level did not lead to protection from PRV-1 and HSMI, the specificity against PRV antigens and association with virus eradication is notable. 

The inactivated, adjuvanted PRV-1 vaccine did not induce any measurable antibodies against PRV-1 σ1. Still, the inactivated PRV-1 vaccine lowered PRV-1 infection levels after secondary challenge, and protected six out of eight individuals from HSMI, in line with previous findings [33]. The mechanism behind this effect is not clear, as neither innate immune activation nor cellular immune activation was revealed through the analyses performed here. We cannot rule out if an early immune activation was triggered by the adjuvant or if antibody-based protection targeting PRV-1 antigens other than PRV-1 σ1 is involved [6]. It is also possible that the inactivation procedure may have changed the structure of the σ1 protein in the inactivated viral particle, as it is in an exposed position in the outer capsid.

The PRV-3 pre-exposure totally blocked PRV-1 infection. Cross-protective antibodies are likely to be one explanation but are most likely not the only one. Several fish had very low levels of detectable antibodies in plasma after 10 weeks, but PRV-1 infection was still completely blocked in these fish. The analysis of expressed antiviral immunity genes and indicators of cellular adaptive immunity (cytotoxic cell markers) did not indicate that these mechanisms were triggered by PRV-3 beyond 10 weeks, at least not in spleen, which was tested. The almost total infection block may lead us to think that protective mechanisms have been induced also at mucosal surfaces, although PRV-3 was given ip and not as a bath exposure. In general, orthoreoviruses enter through respiratory and gastrointestinal mucosal surfaces. Although PRV-1 is reported to infect via the intestinal wall [13], it may use other ports of entry as well. The infection route of PRV-3 has not been studied but could be assumed similar to that of PRV-1. This could point to a mucosal protection mechanism involved in the blockage of PRV-1 infection by PRV-3, which would be a highly desired effect of a future vaccine. Such a PRV-1-blocking effect was not obtained with previous PRV-2 exposure or the inactivated PRV-1 vaccine. It should be noted that PRV-3, but not PRV-2, persisted in the spleen of all fish when they were exposed to PRV-1, and further until the end of the study (18 weeks). It may be that the almost full protection and blocking of PRV-1 infection is dependent on the presence of PRV-3. 

All PRV isoforms infect red blood cells, and PRV-1 is shown to strongly induce interferon-regulated antiviral genes in these cells [50]. Thus, the blocking of secondary PRV-1 infection could be a result of red blood cells in an antiviral state. This would be reflected in analysis of spleen antiviral responses. However, very little innate antiviral immune response was induced by PRV-3 in Atlantic salmon although fish were still infected with the virus after 10 weeks. This is remarkably different to a PRV-1 infection, which induces long-lasting antiviral responses. PRV-3 is also reported earlier to induce antiviral responses in rainbow trout [29,30], but not in Atlantic salmon [29]. This difference could be linked to the observed differences in pathogenicity in the two species, but this is still not confirmed and will be further explored. 

For PRV-2, 50–80% of the fish had cleared the virus between 10 and 18 weeks after infection. In this group, we found a contradictory effect on PRV-1 infection and HSMI. As two out of eight fish did not develop HSMI, there was no effect on the remaining six. In addition, PRV-1 levels were lower in some fish, but strongly boosted in others. It appeared that PRV-1 may have replicated more efficiently in some of the individuals that had eradicated PRV-2, compared to individuals that had not. Like for PRV-3, we did not detect innate antiviral immune responses to PRV-2 infection 10 weeks after infection. 

PRV-1 induces cytotoxic T-cell (CTL) activity in Atlantic salmon [17,37], which is strongly associated with HSMI pathology [16], and also heart inflammation in rainbow trout infected with PRV-3 [30]. Here, there is clear evidence that PRV-1 induces a strong regulation of innate antiviral and cytotoxic immune response genes 10 weeks after infection, which is not induced by PRV-2 or PRV-3, and which is likely to be decisive for HSMI pathology. The role of CTL activity in vaccine effects and long-term protection against viruses in salmonids is not much studied, but specific cytotoxicity against the salmonid alphavirus (SAV) has recently been explored after vaccination with an adjuvanted inactivated SAV vaccine, in comparison with SAV infection [51]. There, it was clearly demonstrated that while SAV infection induced specific cytotoxicity, only unspecific cytotoxic activity was induced by the vaccine [51]. It would, in a follow-up study, be interesting to compare specific CTL activity in the period after PRV-2 and PRV-3 infection to explore any correlation with the ability to cross-protect against PRV-1. 

This study illustrates some potential pitfalls in using replicating viruses for vaccine purposes. They may be very efficient, like PRV-3, which completely blocks PRV-1 infection. However, PRV-3 itself persists in the fish, which may have unknown long-term consequences. 

This study also indicates that antibodies against the putative receptor-binding protein σ1 may be an important protective measure. PRV-3, but not PRV-2, induced the production of anti-σ1 antibodies. This could be due to the higher replication rate of PRV-3 to PRV-2 in Atlantic salmon and the higher identity between the σ1 protein of PRV-1 and -3. A protective effect could eventually be verified in a passive immunization test by administration of purified serum immunoglobulin from PRV-3-infected fish to PRV-1 experimentally infected fish. The long-term protective effects of these antibodies will be subject to follow-up experiments, as we could observe a decline after > 15 weeks of PRV-3 infection. If plasma antibodies are also involved in blocking infection at mucosal surfaces is an open question.

PRV-2 replicates at low levels in Atlantic salmon and is eventually cleared, which normally could be considered beneficial properties of a “live” replicating vaccine. However, the replication must be at a level sufficient to induce an effective immune response. Here, only minor innate and cellular responses were found at the transcript level. However, there may be effects at the post-transcriptional or post-translational levels that we did not monitor. PRV-2 caused contradictory results by protecting some fish from HSMI but causing even higher susceptibility to PRV-1 infection in others. The large individual differences could possibly be due to host genetics, leading to a different ability to present antigen. This study also confirms the partial efficiency of the inactivated PRV-1 vaccine published earlier; although, it is still without a clear answer to the main mechanism of protection. 

Besides the obvious pitfalls in immunizing Atlantic salmon against HSMI with PRV-3, a virus pathogenic to rainbow trout [28], there are also additional concerns associated with a live attenuated vaccination approach. Segmented RNA viruses may reassort or recombine if two related genotypes infect the same cell [52], creating new viruses with unpredictable properties, potentially pathogenic. 

Future vaccine production can provide us with reverse genetic approaches and viruses tailored by synthesis and gene editing. Combined with thorough long-term studies of risks and effects of the different vaccine approaches and a higher repertoire of ways to measure vaccine effect, this will hopefully ensure safe and effective attenuated vaccines in the future.

## 5. Conclusions

This work show that PRV-1, PRV-2 and PRV-3 replicate in Atlantic salmon, and can induce production of antibodies that bind to the PRV-1 σ1 antigen. Only PRV-1 in-fection induce unspecific antibodies, long-lasting expression of antiviral response genes and cytotoxic genes in spleen in Atlantic salmon, which could be associated with the ability to cause HSMI. When compared to vaccination with an inactivated PRV-1 vaccine, PRV-3 infection provides full protection from PRV-1 introduced ten weeks later, and development of HSMI. In comparison, inactivated PRV-1 vaccine and PRV-2 infection does not prevent PRV-1 infection and only partially protects against HSMI. This work indicates that a replicating attenuated vaccine approach could protect against HSMI.

## Figures and Tables

**Figure 1 vaccines-09-00230-f001:**
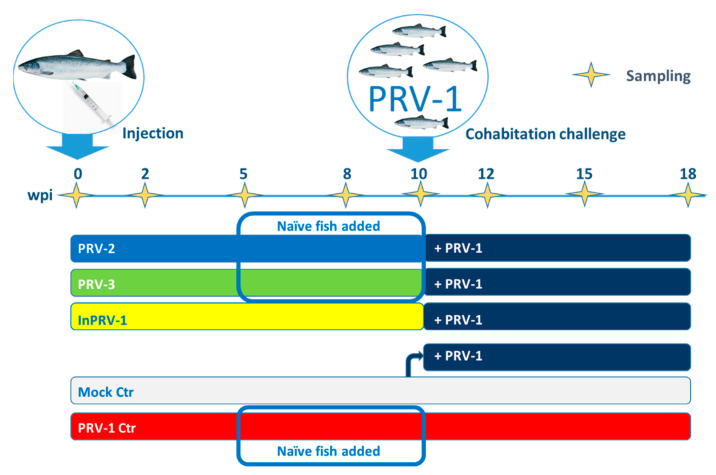
**Groups and timeline of the *Piscine orthoreovirus* (PRV) immunization trial.** Fish were immunized intraperitoneally (ip) with either spleen homogenate containing PRV-2 (blue group), blood cell lysate containing PRV-3 (green group), or purified, inactivated and adjuvanted PRV-1 (InPRV-1, yellow group). The negative control group (mock, white) was injected with blood cell lysate from uninfected fish. A positive control group was injected with PRV-1 (red). Naïve fish were added to tanks containing fish injected with infective PRV-1, PRV-2 and PRV-3 five weeks post injection (wpi) and sampled Week Eight and Ten to control viral shedding. After 10 weeks, the immunized group and half of the mock group was infected through cohabitation with fish experimentally infected with PRV-1 (shedders, dark blue) and thereafter monitored until Week 18. Yellow stars on the timeline show sampling time points (all groups).

**Figure 2 vaccines-09-00230-f002:**
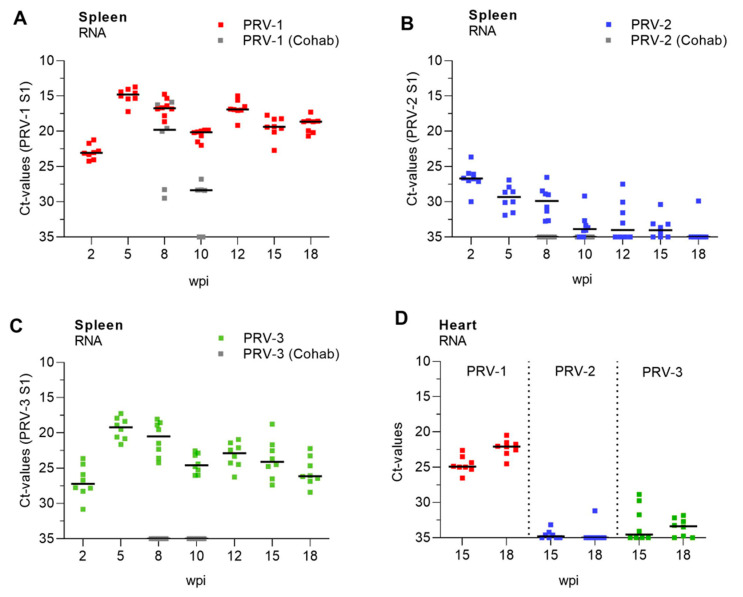
**Development of infection with PRV-1, -2 and -3**. Levels of PRV-1 (**A**), PRV-2 (**B**) and PRV-3 (**C**) as detected in spleen with specific RT-qPCR assays targeting the S1 genome segment in the respective viruses and trial groups. The figures show individual Ct values and median (line) at each sampling from 2 to 18 weeks post injection (wpi). Gray dots show virus levels in naïve cohabitants added to the tank at 5 wpi and removed at 10 wpi (5 weeks after exposure). Relative levels of PRV-1, -2 and -3 in heart at 15 and 18 weeks post infection (**D**).

**Figure 3 vaccines-09-00230-f003:**
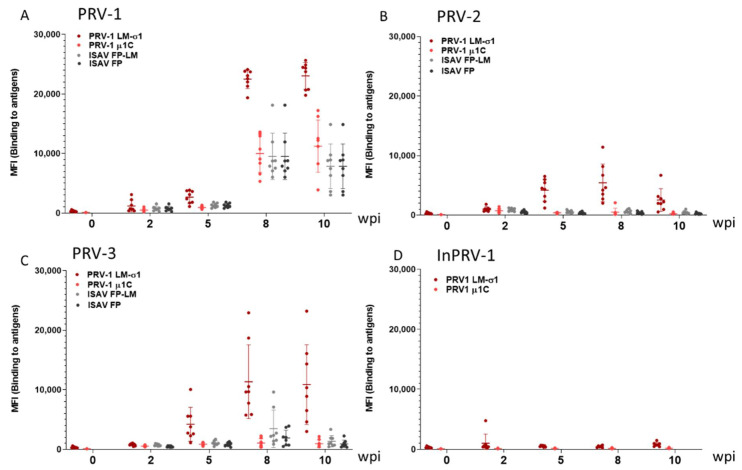
**Production of anti-PRV antibodies.** Magnetic beads coated with recombinant lipid-modified (LM)-PRV-1-σ1, PRV-1 µ1c, infectious salmon anemia virus fusion protein (ISAV-FP) or LM-ISAV-FP in a multiplexed assay were used to measure PRV-specific and unspecific antibodies in blood plasma sampled from fish in the PRV-1 (**A**), PRV-2 (**B**), PRV-3 (**C**) and InactPRV-1 (**D**) injected groups in the first 10 weeks post injection (wpi). MFI: median fluorescence intensity. The results from beads coated with PRV antigens are shown in red, and beads with non-PRV antigens in gray/black.

**Figure 4 vaccines-09-00230-f004:**
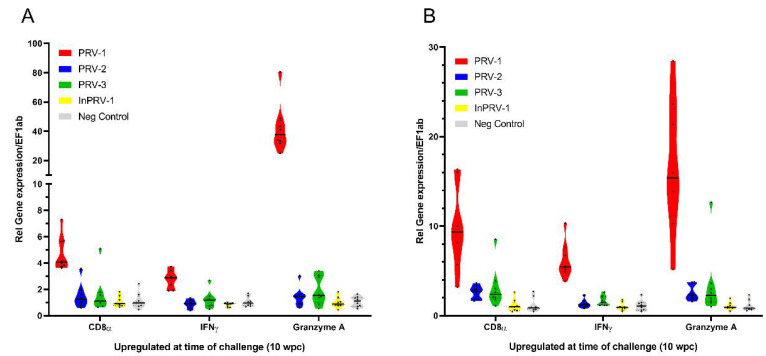
**Cellular and innate antiviral immune responses 10 weeks after immunization (wpi).** Cellular responses CD8α, IFNγ and Granzyme A (**A**) and innate antiviral responses viperin, myxovirus resistance gene (Mx) and interferon-stimulated gene (ISG)15 (**B**) were analyzed by RT-qPCR in spleen, normalized for the reference gene EF1α and shown as 2^−ΔΔCt^ levels. Gene expression in spleen samples from fish injected with PRV-1 (red), PRV-2 (blue), PRV-3 (green), inactivated PRV (yellow) and mock lysate (gray) are shown.

**Figure 5 vaccines-09-00230-f005:**
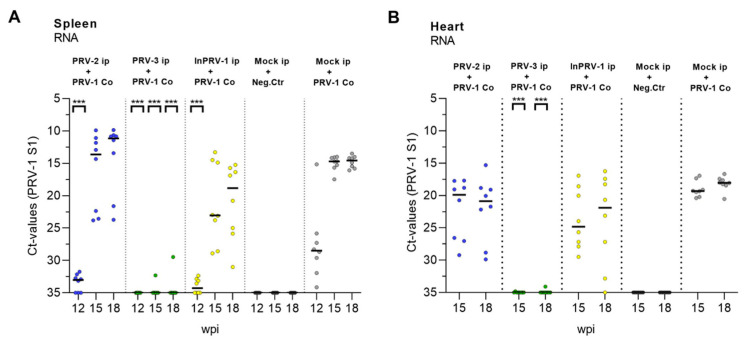
**Development of PRV-1-infection after exposure by cohabitation.** The level of PRV-1 after infection with PRV-1 shedders at 10 weeks was monitored by RT-qPCR at Weeks 12, 15 and 18 in the spleen (**A**) and Weeks 15 and 18 in the heart (**B**). Each dot represents an individual Ct value with a line (median) at each sampling. Dot color: Fish pre-injected with PRV-2 (Blue), PRV-3 (green), Inactivated PRV-1 (Yellow), or mock (grey), then secondary infected with PRV-1 where marked. Statistical analyses were performed by comparing each target group with the PRV-1 control group at each time point using the Mann–Whitney test. Asterisk shows significant difference (*** *p* < 0.001); wpi = weeks post immunization.

**Figure 6 vaccines-09-00230-f006:**
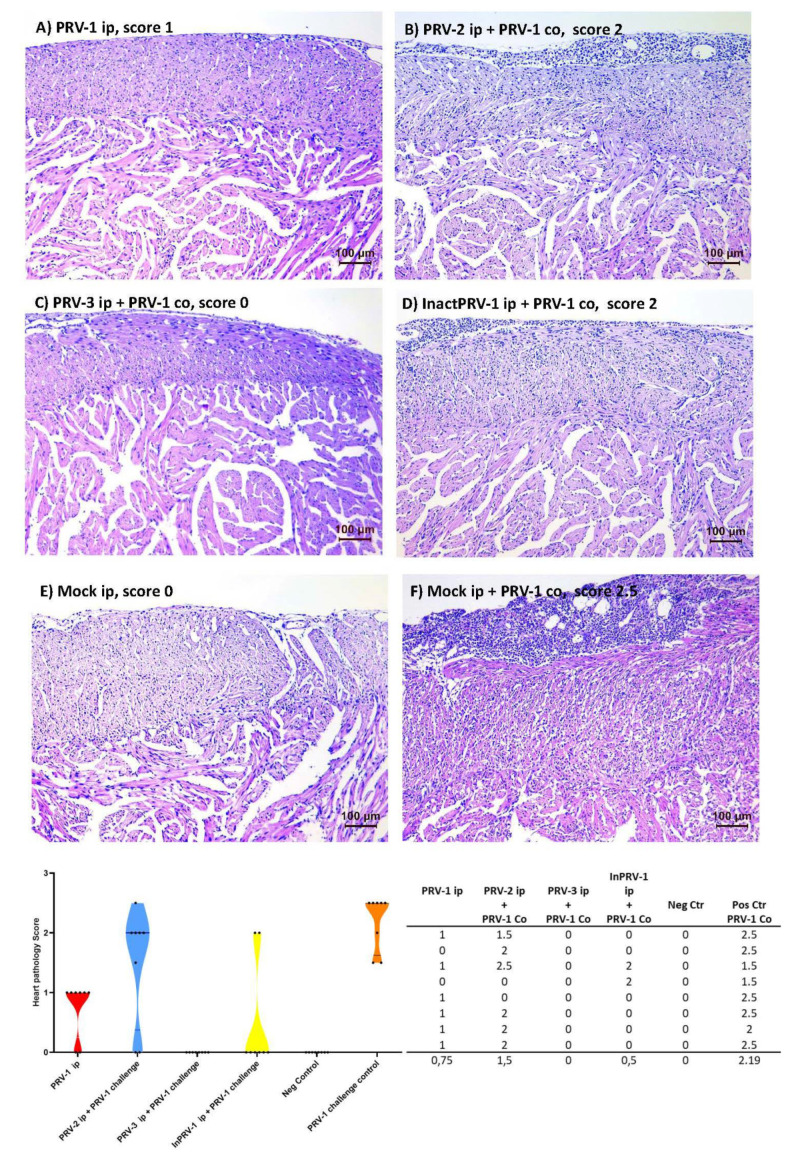
**Histopathology and scores of A. salmon hearts.** The status of Atlantic salmon hearts was scored for heart and skeletal muscle inflammation (HSMI) pathology 18 weeks after immunization and eight weeks after PRV-1 cohabitant challenge. The scoring of pancarditis was performed according to a visual analogue scale from 0 to 3, where 0 represents a healthy heart, scores above 1 represent hearts with increased cellularity due to immune cell recruitment in the outer epicardial layer, and more severe cases (a score of 2,5) also show increased cellularity in the outer compact and inner spongious layers of the heart ventricle. (**A**) PRV-1 ip injected Week 0, (**B**) PRV-2 immunized ip + PRV-1, (**C**) PRV-3 immunized ip + PRV-1, (**D**) inactivated InPRV-1 immunized ip + PRV-1, (**E**) negative control, mock-injected ip, (**F**) positive control, mock-injected ip + PRV-1, and (**G**) a table and violin plot showing pathology scores of individual fish in each experimental group (*n* = 8) pre-injected with PRV-1 (red), PRV-2 (Blue), PRV-3 (green), Inactivated PRV-1 (Yellow), or mock (grey), then secondary infected with PRV-1 where marked.

**Figure 7 vaccines-09-00230-f007:**
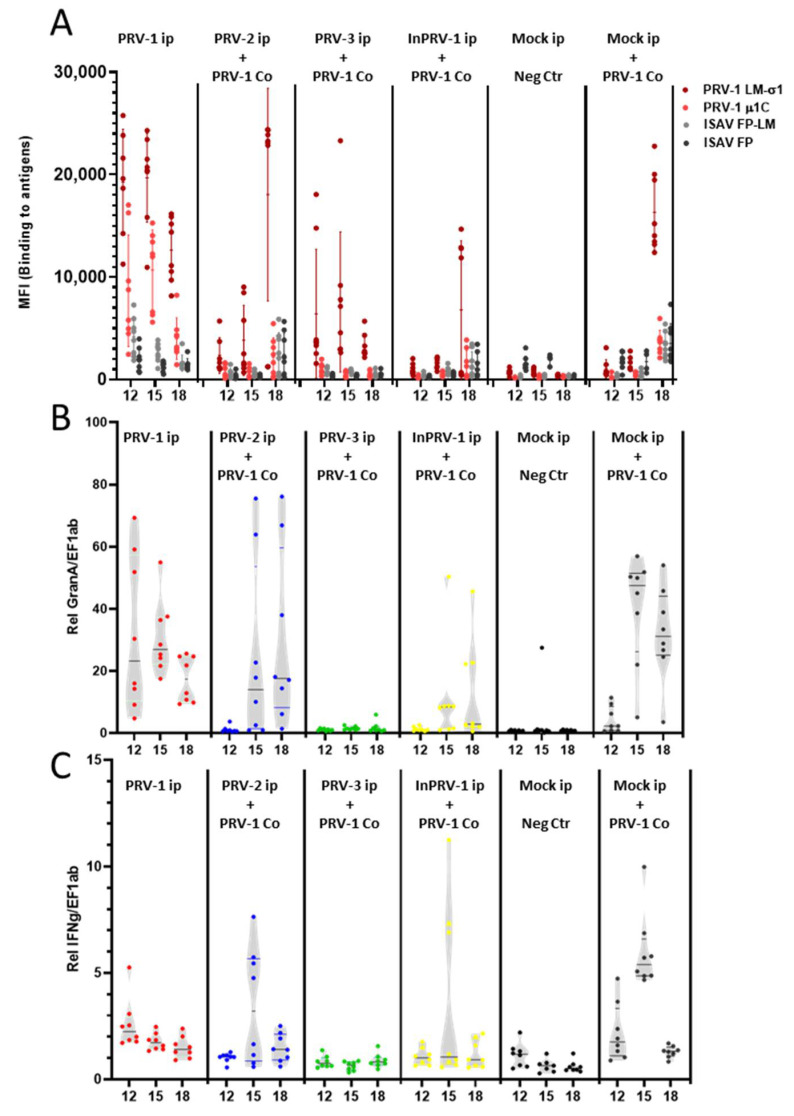
**Immune responses in the spleen after PRV-1 challenge of immunized fish.** (**A**) Magnetic beads coated with recombinant lipid-modified PRV antigens (LM-PRV-1σ1, PRV-1 µ1c), and non-PRV antigens (ISAV-FP or LM-ISAV-FP), used in a multiplexed assay to measure antibodies from diluted blood plasma sampled from fish in the trial groups. Levels of fluorescent secondary antibody bound to the beads (median fluorescence units, MFI) carrying PRV-antigens (red) and non-PRV antigens (gray/black) were assayed. (**B**) Gene expression of Granzyme A. (**C**) Gene expression of IFNγ in spleen samples from fish injected with PRV-1 (red), PRV-2 + PRV-1 (blue), PRV-3 + PRV-1 (green), InactPRV-1 + PRV-1 (yellow), mock negative control, and mock + PRV-1 positive control groups (black).

## Data Availability

The supplementary data files presented in this study are openly available in Zenodo, at https://doi.org/10.5281/zenodo.4450287 (accessed on 6 March 2021).

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
