# Peer review of "Piscine Orthoreovirus (PRV)-3, but Not PRV-2, Cross-Protects against PRV-1 and Heart and Skeletal Muscle Inflammation in Atlantic Salmon"

_vaccines, 2021, doi:10.3390/vaccines9030230_

Round 1

Reviewer 1 Report

The manuscript by Muhammad Salman Malik and colleagues presents and original work about the cross-reactive protection of PRV-2 or PRV-3 infection in Atlantic salmon could provide against a consecutive PRV-1 infection and HSMI. They also compared the protection with an inactivated PRV-1 vaccine. The results show that PRV-3 infection in Atlantic salmon, in contrast to PRV-2, blocks a secondary infection with PRV-1, and that cross-protective antibodies may be one of the mechanism involved.

I found this article really interesting, properly structured and performed and well written. I only have some questions and minor comments for the authors.

Questions

  • In this manuscript to analyzed the immune gene expression authors chose spleen as a target tissue. I was wondering why they did not study the head kidney since it is also an important tissue in the immune response.
  • In figure 2 it is clear that fish infected with PRV-2 and PRV-3 reduce their infection from week 10 to 18. My question is why authors evaluated the infection in heart at 15 and 18 wpi instead at early wpi where the infection is higher in spleen?
  • In figure 4B authors analyzed the immune response at 10 wpi. I believe that at 10 wpi it is a bit late to analyzed the innate immune response. Have authors try to see the innate immune response at earlier weeks?
  • The results obtained in this manuscript show that PRV-3 infection in Atlantic salmon, blocks a secondary infection with PRV-1. I was wondering if authors have studied if RBCs played some kind of role in this cross-protection since RBCs are the target from PRV-3?.

Minor comments

Line 51-53 sentence "In Atlantic salmon aquaculture, vaccines have been effective in protecting the fish from many diseases, but several viral diseases remain unsolved challenges". This sentence needs a reference, please add it.

Line 78 "has been shown to to differ". Please remove one to.

Line 155 Please add commercial reference of heparin-coated vacutainer tubes.

Line 156 Please add the place of Qiagen at least the first time that it is mentioned in the manuscript as the rest of commercial references.

Line 172 Agilent Technologies (Texas, USA), please correct it to (Agilent Technologies, Texas, USA).

Line 187 Please add the place of (Bio-Rad) at least the first time that it is mentioned in the manuscript.

Line 205 Please remove the gap in ".   All incuba"

Line 209 Please define BSA at least the first time that it is mentioned in the manuscript.

Line 229 Please add commercial reference of (H&E).

Line 230 Please remove the gap in " croscopy.   The ".

Line 236 Please define WPC.

Line 259 Please define wpi, in the legend of figure 1.

Line 268 Please define IQR at least the first time that it is mentioned in the manuscript.

Line 287 wpi and 288 WPI is in capital letter. Please unify this in all the manuscript.

Figure 3and 4 legend a)b) c ) and d) are not in capital letters as the rest of legends. Please change them.

Line 316 Please define" i.e. ".

Line 319 Please define" Mx and ISG15. ".

Line 337 and 345 Please remove the gap.

In Figure 6 I found difficult to evaluate the histopathology because I am not familiar with this technique. As a piece of advice for the next readers, you should try to explain a bit the difference between 0 score and 2.5 in the images.

Table S1. Primers of immune genes sequences should include reference or accession number.

Author Response

Response to reviewer 1

We thank the reviewer for comments and suggestions to help improve our manuscript.  

We have the following response and have made the following revisions 

1: In this manuscript to analyze the immune gene expression authors chose spleen as a target tissue. I was wondering why they did not study the head kidney since it is also an important tissue in the immune response.

Response:

In this study we originally chose to analyse spleen for levels of PRV. Red blood cells are the main target cells of all PRV genotypes, and we knew that levels in spleen reflected blood cell levels better than head kidney. We know from earlier that red blood cells infected with PRV-1 respond with a strong innate antiviral response which is also measured in spleen, and that induction of genes reflecting cellular immunity (cytotoxic T-cells ) can be measured in spleen as well, whereas genes associated with B-cell immunity are more strongly induced in kidney (Johansen et al 2016), as the reviewer correctly points out. Since we had monitored antibody production in this study, we felt we had a grip on the state of humoral (B-cell) immunity, and our main aim was to look into innate antiviral and cellular immune response, for which spleen is a better target organ.  It should be noted that we based our choices on the knowledge we have on PRV-1, aware that there are many unknowns regarding PRV-2 and PRV-3-mediated infection and immunity in Atlantic salmon.

We have now more clearly addressed why we chose spleen for these analyses in the manuscript (L270-272 and L321-322).

In figure 2 it is clear that fish infected with PRV-2 and PRV-3 reduce their infection from week 10 to 18. My question is why authors evaluated the infection in heart at 15 and 18 wpi instead at early wpi where the infection is higher in spleen

Reply: In this study, we did not focus on exploring the spread of PRV-2 and PRV-3 to the Atlantic salmon heart at the peak of infection, but on cross-immunity and protective effects on PRV-1 mediated HSMI pathology. For that reason, hearts were only analysed at the end of the study, to ensure that PRV-1 was dominating in the infected hearts, and the actual cause of pathology. We have previously reported that PRV-3 infection leads to minor changes in A. salmon hearts (Hauge et al. 2017 ). We can not from these data say if PRV-2 and PRV-3 caused minor heart pathology at an earlier time during this study, just that there were no signs of pathology and low virus levels at the end.We agree this should be further studied.

This limitation of the study on PRV-2 and PRV-3-mediated pathology in other tissues/other time-points is now clarified in the discussion (457-459). We also clarify the purpose in results (L294-295)

In figure 4B authors analyzed the immune response at 10 wpi. I believe that at 10 wpi it is a bit late to analyzed the innate immune response. Have authors try to see the innate immune response at earlier weeks?

Response: PRV-1 mediated innate immunity have previously been studied at earlier time points after PRV-1 and PRV-3 infection in Atlantic salmon (Wessel, 2017 Johansen, 2016, Vendramin 2019 ). The reason for studying immune responses at 10 wpi here (immediately prior to exposure to PRV-1 shedder fish) was to be able to associate immune response data more closely to protection against PRV-1.

The purpose is now clarified (L321-322)

The results obtained in this manuscript show that PRV-3 infection in Atlantic salmon, blocks a secondary infection with PRV-1. I was wondering if authors have studied if RBCs played some kind of role in this cross-protection since RBCs are the target from PRV-3?.

Response:  We have not explored  in this trial if PRV-3 mediated infection of RBC directly protects from PRV-1 infection, but this could be an explanation and is obviously interesting to explore further. The ability of different PRV genotypes to infect the same cell is interesting for many reasons, Both for understanding antiviral immunity and for the possibility of reassortment between the two viruses.  However, we observe here that antiviral immune responses induced by PRV-3 infection (at the transcriptional level) is low compared to PRV-1.

 We have now addressed this point in the discussion (511-513, and 551-555)

All minor comments are addressed directly in the manuscript and marked in the file with revisions. Se attached revised file.

Reviewer 2 Report

Dear Ms. Lia Liang

E-Mail: [email protected]

MDPI Tianjin Office 170 North Road, Room 1804, Block A, Lujiazui Financial Plaza, Hongqiao District, China

Reviewer comments : Manuscript ID: vaccines-1105449 – submitted to Vaccines

Title: Piscine orthoreovirus (PRV)-3, but not PRV-2, cross-protects against PRV-1 and heart and skeletal muscle inflammation in Atlantic salmon

This manuscript looks interesting, the scientific approach of the work is rigorous, complete and well developed.

It will certainly be an excellent contribution to better understand the the role of non-pathogenic PRV in a future to mitigate the spread of HSMI in salmon farming. The theme falls within the broader theme of the study of vaccines and antibody protection, very current and in line with the so-called “food security”.

The bibliography is also complete and updated. 

I express some advice that are not to be considered as a revision, but as a contribution to improve the work done:

In the “Discussion” it would be useful to mention the fact that PRV-1 is still the subject of genetic studies that are highlighting some characteristics such as Genetic diversity (Siah et al., 2020), Sub-genotypes presence (Godoy et al., 2021),

These peculiarities must be considered as they can affect the interfering action of PRV-3, because they are effective on some strains of PRV-1 but perhaps not on all.

  1. Siah et al. , 2020, Genomes reveal genetic diversity of Piscine orthoreovirus in farmed and free-ranging salmonids from Canada and USA, Virus Evolution, 6(2): veaa054, doi: 10.1093/ve/veaa054

Marcos Godoy et al., 2021, Extensive Phylogenetic Analysis of Piscine Orthoreovirus Genomic Sequences Shows the Robustness of Subgenotype Classification Pathogens, 10, 41; https://doi.org/10.3390/pathogens10010041

Again in the Discussion, the fact of the application in the practice of breeding of this cross-coverage of PRV-3 to PRV-1 deserves a mention. Indeed, the mutability and species jumping ability of orthoreoviruses requires caution on this aspect. See also Polinski et al, 2020, Piscine orthoreovirus: Biology and distribution in farmed and wild fish, J Fish Dis. 2020;43:1331–1352. https://doi.org/10.1111/jfd.13228

There are quite minor revisions

Minor revisions:

Line 74: ..black spot….. the role of PRV-1 in this partology seems really marginal, while a nutritional cause is much more plausible. See also Turid Mørkøre et al., 2018, Nutritional effects on dark fillet spots of Atlantic salmon (Salmo salar L.), 18th International Symposium on Fish Nutrition and Feeding Las Palmas de Gran Canaria, Spain June 3rd – 7th,

Line 117..ct values……moves in  qPCR ct values

Line 137… a parenthesis has been closed, but you can't see where it was opened

References:

line 594: even if not all the original titles listed in the bibliography mention the binomial name of the species in italics, it would be better to standardize and report them all in italics.

So for example, in this line, change Salmo salar in Salmo salar..

Please check the other bibliographic entries

Author Response

Response to reviewer 2

We thank the reviewer for comments and suggestions to help improve our manuscript. 

We have the following response and have made the following revisions 

I express some advice that are not to be considered as a revision, but as a contribution to improve the work done: In the “Discussion” it would be useful to mention the fact that PRV-1 is still the subject of genetic studies that are highlighting some characteristics such as Genetic diversity (Siah et al., 2020), Sub-genotypes presence (Godoy et al., 2021). These peculiarities must be considered as they can affect the interfering action of PRV-3, because they are effective on some strains of PRV-1 but perhaps not on all.

 Response: We agree that PRV-1 diversity this is an important point that have not been adressed strongly in this work. We have now added a section in the discussion, and included the two suggested references (and another two on this topic) L444-447

 Again in the Discussion, the fact of the application in the practice of breeding of this cross-coverage of PRV-3 to PRV-1 deserves a mention. Indeed, the mutability and species jumping ability of orthoreoviruses requires caution on this aspect. See also Polinski et al, 2020, Piscine orthoreovirus: Biology and distribution in farmed and wild fish, J Fish Dis. 2020;43:1331–1352. https://doi.org/10.1111/jfd.13228

Response: We agree that there are many concerns associated with the ability of PRV variants to cross infect different species. In this manuscript we have chosen not to dive heavily into this area, and keep a focus on the cross protection ability in a vaccination setting. We have addressed this, and also the reassortment aspect in the discussion (L430-431, L551-555). We have also referred to the review in the introduction (L 81-82).

  Line 74: ..black spot….. the role of PRV-1 in this partology seems really marginal, while a nutritional cause is much more plausible. See also Turid Mørkøre et al., 2018, Nutritional effects on dark fillet spots of Atlantic salmon (Salmo salar L.), 18th International Symposium on Fish Nutrition and Feeding Las Palmas de Gran Canaria, Spain June 3rd – 7th,

Response: We agree the link between PRV-1 and black spots is not clearcut and direct, and have added a sentence that the PRV-Black spot link is disputed (L76-77). It should be mentioned that we have data in the pipeline indicating that PRV-1 infection may worsen spot formation due to direct effects on the local inflammatory response.

The minor comments  are addressed directly in the manuscript with revisions
